# Sustenance Trial to Analyze the Effects of Black Soldier Fly Larvae Meal on the Reproductive Efficiency of Sows and the Hematological Properties of Suckling and Weaning Piglets

**DOI:** 10.3390/ani13213410

**Published:** 2023-11-03

**Authors:** Kiyonori Kawasaki, Junliang Zhao, Natsu Takao, Masaki Sato, Takuma Ban, Kaoru Tamamaki, Masanori Kagami, Kiminobu Yano

**Affiliations:** 1Department of Applied Biological Science, Faculty of Agriculture, Kagawa University, Ikenobe 2393, Miki-cho, Kita-gun, Kagawa 761-0795, Japan; cho.syunryo@kagawa-u.ac.jp (J.Z.);; 2University Farm, Kagawa University, Showa 300-2, Sanuki, Kagawa 769-2304, Japan

**Keywords:** *Helmetia illucens*, black soldier fly, reproductive efficiency, litter size, nursing piglets, weaning piglets, insect meal

## Abstract

**Simple Summary:**

The escalating demand for meat, propelled by a worldwide surge in population and economic development, necessitates the implementation of sustainable approaches for animal feed production. This study investigated the impact of incorporating black soldier fly (BSF; scientific name: *Hermetia illucens*) meal into the diets of sows and piglets as a potential substitute for conventional protein sources on sow fertility, blood parameters, piglet growth, and intestinal tissue structure. Consuming BSF instead of animal or soybean proteins did not deteriorate sow reproductive performance. The feed conversion ratio from 28- to 35-day-old piglets was lower in BSF-fed groups despite no discernible disparities in growth; this indicated that it may be beneficial to optimize the rate of replacing animal proteins with BSF meal, preferably along with chitin removal from BSF meal to improve digestibility. Furthermore, minor variations in blood cell composition and properties were observed in 28-day-old piglets, with elevated high-density lipoprotein cholesterol levels in the high-BSF group, which may be attributed to the unique fatty acid composition of BSF.

**Abstract:**

The escalating demand for meat, driven by global population growth, necessitates sustainable solutions for animal feed production. This study investigated the effects of substituting conventional protein resources in sow and piglet dietary regimens with black soldier fly (BSF; *Hermetia illucens*) meal on reproductive efficiency, blood profile, piglet growth, and intestinal tissue morphology. The results indicate that substituting animal-derived and soy proteins with BSF meal does not compromise sow reproductive performance. Although no notable disparities were observed in piglet growth, the feed conversion ratio from the 28- to 35-day age marks were lower in the BSF-fed groups. This suggests that the animal protein-BSF substitution rate may require optimization, potentially involving chitin removal from BSF meal to enhance digestibility. Minor variations in the hematological composition and properties in piglets, with elevated high-density lipoprotein cholesterol levels in the high BSF group at the 28-day mark, were potentially attributable to the unique fatty acid composition of BSF meal. Moreover, this study potentiates future exploration into the efficacy of complete animal protein substitution with BSF meals on piglet nutrition and physiology, particularly in fattening pigs. The practical implementation of BSF meals in animal feed production holds promise for enhancing the sustainability of the swine industry.

## 1. Introduction

The recent increase in meat consumption is attributable to the worldwide surge in population growth and economic development in developing countries. Specifically, estimates have projected that pork production will need to be augmented by approximately 40% by 2050 compared with the levels recorded in 2010 [1]. Pigs are omnivorous mammals that predominantly subsist on a diet comprising grains and vegetable oil cakes. Additionally, they are supplemented with animal proteins, such as fish and chicken meals, owing to their superior amino acid profiles [2,3]. Fishmeal, in particular, raises sustainability concerns because it is heavily contingent upon natural resources, susceptible to price fluctuations, and potentially competes with food resources allocated for human consumption [4]. Furthermore, soybean meal, which serves as a significant component of vegetable oil cakes used in animal feeds [5], is often considered suitable owing to its utility as a by-product of soybean oil. However, a majority of the global soybean production, approximately 80%, is allocated for livestock feed, while a mere 20% is designated for human consumption [6]. Additionally, notwithstanding the substantial increase in soybean production over the past decade, a notable fraction of it has been subjected to deforestation in South America to acquire land [7]. Consequently, expanding soybean production to meet escalating meat demands would be detrimental to the environment and render soybean production unsustainable.

Given these circumstances, insects have emerged as a novel source of sustenance in recent years [8,9]. Particularly, the black soldier fly (BSF; *Hermetia illucens*), has garnered considerable attention for its remarkable efficiency in converting organic waste into animal proteins [10]. Extensive research has been focused on exploring its suitability as a source of nourishment for fish and poultry [11]. However, there is limited documentation on BSF feeding in pigs, with most reports focusing on short-term trials [11,12]. To the best of our knowledge, no prior studies have investigated BSF feeding across various stages of growth.

In short-term pig-feeding trials, piglets were provided with BSF meals as a substitute for soybean meal for 15, 21, or 61 days post-weaning [13,14,15]. The results consistently indicated the absence of any statistically significant differences in piglet growth between treatments. The highest BSF inclusion rate among all the trials was determined to be 21% [13]. Additionally, when feeding late-fattening pigs, soybean meal was replaced with 4% or 8% BSF, which markedly enhanced growth [16]. Nevertheless, the extant literature presents a dearth of any long-term BSF feeding trials, lasting approximately 140 days from sow insemination to piglet weaning.

Recent research has evidenced that incorporating fiber sources into sow diets can improve their intestinal health and reproductive performances [17,18]. Unlike conventional animal proteins, insect powders are enriched with chitin, which is a constituent of insect shells [19]. Chitin transforms various animal intestines, yielding short-chain fatty acids, namely, acetic and propionic acids [20,21,22,23]. A previous study demonstrated that including 4% BSF in the diet substantially altered the intestinal microbiota and metabolites of fattening pigs, ultimately enhancing the mucosal immune balance [21]. Consequently, this outcome prompted the hypothesis that supplementing sow diets with BSF meals could potentially improve reproductive performance through the optimization of the intestinal environments in sows.

Additionally, piglets encounter a multitude of stressors during the weaning process, including transitioning from milk to solid sources of nourishment and cohabiting with piglets from different litters in unfamiliar surroundings [24]. These stress-inducing factors have the potential to abruptly halt the consumption of feed after weaning [24,25], resulting in diminished body weight gain and substantial economic ramifications for swine breeders. To address this issue, pre-weaning diets are given to piglets [26]. However, these dietary regimens often contain animal-derived proteins, some of which compete directly with essential human food resources, such as whey protein and skimmed milk powder [27]. Substituting these animal proteins in BSF meals could potentially enhance the sustainability of the swine industry.

Therefore, this study aimed to comprehensively investigate the impact of substituting fish meal with BSF meal in sow diets, starting from the day of insemination until the weaning of their nursery piglets. Additionally, this research evaluated the effects of substituting animal-based proteins in the diets of suckling and weanling piglets with BSF meal on piglet growth.

## 2. Materials and Methods

This study was conducted following the review and approval of the Kagawa University Animal Experiment Committee (KU-21671 and KU-22664).

### 2.1. Animals, Diets, and Experimental Procedure

This study involved the establishment of three experimental groups: a control group (C) and groups supplemented with low (L) or high concentrations of BSF (H). Table 1 presents the dietary formulations of the sow and the pre-weaning and post-weaning diets for each experimental group. Table 2 documents the nutrient compositions of the individual diets and the BSF meal (Entofood, Kuala Lumpur, Malaysia) utilized in the present study. BSF larvae were reared by feeding local vegetable waste. The devitalization of BSF larvae was conducted using a hot bath. After the devitalization, the BSF larvae were minced and defatted using a centrifuge tricanter, and then solid material was dried at 80 °C for 5 h.

A total of thirty female pigs (Landrace × Large white; LW), including both gilts and sows, were allocated to each group (Appendix A). Artificial insemination was used for siring all females, utilizing Duroc (D) boar semen. The farrowed piglets (LWD) were accommodated in the same stall as their sows and were provided unrestricted access to milk and water until weaning. The male piglets were castrated at the age of 21 days and supplemented with a daily pre-weaning diet of 100 g per piglet for a duration of 7 days until they reached the age of 28 days. Throughout the designated research period, the sows were sustained on a daily ration of 2.5 kg of the prescribed diets in the mornings (1.25 kg) and evenings (1.25 kg) during their gestation periods. Subsequently, the daily provision was increased to 6.0 kg of the recommended diet in the morning (3.0 kg) and evening (3.0 kg) during suckling. Water was provided ad libitum. The piglets were weighed weekly during their suckling phase, and their growth performance and survival rates were recorded meticulously up until they reached 28 days of age. Upon reaching 28 days of age, the piglets were weaned. Individuals whose weights approximated the average weight of the piglets in each litter were selected and randomized in five pens, with thirty piglets per experimental group (Appendix A). These animals were then fed the specified pre-weaning diets from 21 to 35 days of age, followed by the post-weaning diets from 35 to 63 days of age.

In the sow diets, the L group contained 2.3% BSF meal, and the H group had 4.6%. The L group substituted half of the fishmeal with BSF meal, while the H group completely substituted fishmeal with BSF meal. Soybean meal in the diets was reduced by 6.0% in the L group compared with the C group and by 29.0% in the H group compared with the C group. In the pre-weaning diets, the L group contained 12.5% BSF meal, and the H group had 25.0%. The substitution of animal protein and BSF meal in the diets were same as the sow diets. Soybean meal in the diets was reduced by 16.0% in the L group compared with the C group and by 44.0% in the H group compared with the C group. In the post-weaning diets, the L group contained 4.0% BSF meal, and the H group had 8.0%. The substitution of fishmeal and skimmed milk powder with BSF meal was the same as the sow and pre-weaning diets. Soybean meal in the diets was reduced by 5.5% in the L group compared with the C group and by 11.4% in the H group compared with the C group.

Proximal and chemical compositions of the BSF meal and diets were analyzed according to the methods of the Association of Official Analytical Chemists [28]. Moreover, crude protein content was calculated from our analyzed data using a nitrogen-to-protein conversion factor (Kp) of 5.60 for BSF meal content and a Kp of 6.25 for non-insect ingredient content [29,30,31]. The chitin concentration in the BSF meal was determined using the following procedure: 10 g of the BSF meal samples were subjected to reflux in 100 milliliters of 3% sodium hypochlorite solution (v:v) at a temperature of 100 °C for 10 min. Subsequently, the samples were washed with distilled water until the sample solution reached a pH of 7.0, and this process was repeated. Following this, the samples were subjected to reflux in 50 milliliters of 1 M hydrogen chloride solution at a temperature of 75 °C for 15 min. The samples were washed with distilled water until the sample solution reached a pH of 7.0. In addition, the samples were refluxed in 50 milliliters of 1 M sodium hydroxide solution at a temperature of 100 °C for 20 min, and the samples were washed with distilled water until the sample solution reached a pH of 7.0. Finally, the extracts were filtered using FibreBags S (C. Gerhardt Analytical Systems, Bonn, Germany), and the residues were placed in a dry oven at a temperature of 60 °C for 24 h. The dried white residues were regarded as chitin. These analyses were conducted in triplicate.

### 2.2. Sample Collection

Blood samples were obtained from the sows via jugular vein puncture between 9:00 and 10:00 a.m. on three specific occasions: two months after insemination, two weeks prior to farrowing, and after weaning. Approximately 5 mL of venous blood was collected in vacuum blood collection tubes containing EDTA-2Na and those containing heparin. The EDTA-2Na-treated blood sample was promptly cooled on ice and analyzed on the same day using a blood cell analyzer (Thinka CB-1010; Arkray, Kyoto, Japan). The heparin-treated blood specimen was subjected to centrifugation at 3000× *g* for 10 min, and the plasma was stored at −80 °C for subsequent analysis. Plasma samples were thawed at 4 °C and assayed for blood glucose, total cholesterol (T-Cho), high-density lipoprotein cholesterol (HDL-c), and total protein using a blood analyzer (EZ SP-4430, Arkray, Kyoto, Japan). The low-density lipoprotein cholesterol levels were calculated by subtracting HDL-c from T-Cho.

A single 28-day-old piglet with an average body weight was selected from each litter and euthanized by administering sodium pentobarbital (60 mg/kg) through the auricular vein. Approximately 5 mL of portal vein blood was collected in EDTA-2Na-containing and heparin-containing vacuum blood collection tubes. Upon reaching 63 days of age, three piglets from each pen, with weights approximating the average body weight, were subjected to the same aforementioned blood collection procedure. Subsequently, the blood processing methods, as applied to the sows, were replicated. Furthermore, the liver, spleen, and kidneys of the piglets were weighed, and the small intestine length was measured at both the 28-day and 63-day age marks. Duodenum, jejunum, and ileum tissue samples were excised from the small intestine to assess small intestinal tissue, following the protocol outlined by Tsukahara et al. [32]. These tissue specimens were fixed in a 10% formalin solution for 24 h and then transferred to a 70% ethanol solution for tissue preparation and analysis, following the method delineated by Kawasaki et al. [33].

### 2.3. Statistical Analysis

The statistical analysis of all experimental data was conducted using R (version 4.3.1) [34]. All data were analyzed using the Shapiro–Wilk and Levene’s tests for normal distribution and homogeneity of variance. The results concerning the birth litter sizes of the sows were analyzed using a one-way analysis of covariance (ANCOVA), which included parity numbers as a covariate. Other parameters related to sow reproductive performance, hematological composition, and blood properties were analyzed using an ANCOVA, with parity numbers and birth litter size as covariates. Moreover, the growth performance of the piglets was analyzed using an ANCOVA, with piglet body weight at 28 d and sex as covariates. Additionally, the organ weights, hematological composition, and blood properties of piglets were analyzed using an ANCOVA using piglet sex as a covariate. Furthermore, the morphological features of the small intestines of piglets were analyzed using an ANCOVA, with piglet body weight and sex serving as covariates. The Mann–Whitney’s U test (Bonferroni correction) was used to conduct multiple comparisons for the parameters that exhibited a statistically significant difference as determined with the ANCOVA.

## 3. Results

### 3.1. Reproductive Performance of the Sows

Table 3 displays the reproductive performances of the sows. In each of the groups, the total number of farrowing piglets was as follows: 99 in the C group, 88 in the L group, and 123 in the H group. Within the first week after birth, the number of piglets that succumbed was 9 in the C group, 9 in the L group, and 14 in the H group. The primary cause of death for all these piglets during the initial week after birth was crushing. Although no notable disparities in any of the parameters were observed among the groups, the numbers of farrowing piglets and piglets at weaning were highest in the H group and lowest in the L group. Conversely, the H and L groups exhibited the lowest and highest body weights at weaning and body weight gains, respectively. Additionally, no statistically significant difference was recorded in the weaning rates among the groups.

### 3.2. Piglet Growth Performance

Table 4 presents the data pertaining to the weight gain, feed intake, and feed conversion rate of the piglets in each group. No discernible variations were observed among the groups for any of the measured parameters; however, body weight gain during the growth period from 28 to 35 days of age was comparatively lower in the BSF-fed groups than in the C group, whereas the feed conversion ratio was higher in the BSF-fed groups compared with the C group.

### 3.3. Hematological Composition and Blood Properties of Sows

Table 5 outlines the blood cell composition of the sows. Two months after insemination, the lymphocyte counts in the H group were substantially lower than those in the L group. Nonetheless, no marked disparities were observed in the remaining parameters among the two groups. Additionally, 2 weeks prior to farrowing, the platelet distribution width (PDW) in the L group was considerably lower than that in the C group; however, no statistically significant differences were observed in the other parameters among the groups. Furthermore, at the weaning phase of the suckling piglets, no notable variations were observed in the blood cell composition among the groups.

Table 6 depicts the blood properties of the sows. An ANCOVA performed 2 months after insemination revealed a statistically significant difference in albumin levels; nonetheless, multiple comparisons of the groups did not portray any notable distinction. Contrarily, albumin levels at weaning were substantially lower in the H group than in the L group.

Summarily, there were no marked disparities among the groups for any of the parameters, except albumin, at both the 2-month mark after insemination and weaning. Additionally, no statistically significant differences were observed among the groups for any of the parameters measured 2 weeks prior to farrowing.

### 3.4. Piglets Anatomical Parameters

Table 7 documents the organ weights and digestive tract lengths of piglets at two distinctive ages, that is, 28 and 63 days. At the 28-day age mark, the organ weights of the piglets were the highest in the L group and the lowest in the H group, indicating a statistically significant difference between the two cohorts. The organ weights of the 63-day-old piglets were also highest in the L group and lowest in the H group; however, there was no significant difference between them. Additionally, kidney weight per piglet was considerably lower in the H group compared with the C and L groups. No notable disparities were observed between the groups for any of the other parameters or in any of the parameters at the 63-day mark.

Table 8 displays the morphological characteristics of the small intestine of 28- and 63-day-old piglets. At the 28-day mark, the crypt depths in the duodenum were substantially greater in the L and H groups than in the C group. No statistically significant differences were observed among the groups for any of the other parameters. Although an ANCOVA revealed no discernible variation in 63-day-old piglets, group H exhibited considerably higher jejunal villus heights than the C group. Conversely, an ANCOVA indicated a statistically significant difference in ileal villus height; however, this disparity was not corroborated by the results of multiple comparison analyses.

### 3.5. Hematological Composition and Blood Properties of Piglets

Table 9 depicts the blood cell compositions of 28- and 63-day-old piglets. At the 28-day mark, the ANCOVA outcomes revealed significant differences in platelet counts, but those of multiple comparisons did not portray any notable disparities among the groups. Additionally, no statistically significant differences were observed among the groups for any of the other parameters. At the 63-day mark, the mean corpuscular volume (MCV) and mean corpuscular height (MCH) were markedly higher in the H group compared with the L group.

Table 10 documents the blood properties of 28- and 63-day-old piglets. At the 28-day age mark, the H group displayed the greatest elevation in HDL-c levels; nevertheless, no notable variation was observed among the groups for any of the parameters. Additionally, for the piglets at 63 days of age, there were no significant differences among the groups for any of the items.

## 4. Discussion

In this study, we fed sows diets containing fish meal, soybean meal, and soybean oil with BSF meal for 142 d, starting from the time the sows were inseminated until their suckling piglets were weaned. We investigated the effects on reproductive performance, blood cell composition, and blood properties of sows and weaning piglets, as well as on the small intestinal tissues of weaning piglets. Additionally, we raised weaned piglets for 5 weeks until they reached 63 days of age to evaluate the effects of substituting animal protein with BSF meal in piglet diets before and after weaning on piglet growth performance, blood cell composition, blood properties, and small intestinal tissues.

Although there were no significant differences in sow reproductive performance for any of the parameters, birth litter size and weaning litter size were the highest in the H group and the lowest in the L group. These results were inconsistent; however, it is suggested that feeding sows BSF meal does not adversely affect their reproductive performance. Furthermore, post-weaning piglet mortality has been reported to be in the 3–5% range [35], and no piglets died after weaning in this study. This may be due to the limited sample size of 30 piglets per group. It is suggested that more field-specific large-scale rearing experiments are needed to promote the use of BSF meal in swine production.

In the sow diets, we completely substituted fish meal and soybean oil with BSF meal and replaced 30% of the soybean meal with BSF meal. This led to a 0.4% increase in the crude fiber content of the diet. Previous research has shown that improving sow reproductive performance by increasing the crude fiber content requires at least a 3.5% increase from 2.9% to 6.6% [17]. However, in the present study, the increase in dietary crude fiber content due to the substitution of BSF meal with protein resources was not sufficient to affect sow reproductive performance. Moreover, although a previous study suggested that 4% BSF meal feeding improved the finishing pigs’ intestinal environment [21], the diets in this study might not significantly alter the intestinal environment of the sows. Furthermore, BSF meal, an animal protein rich in amino acids, did not significantly affect the total protein content of sow blood, which is an indicator of protein digestion and absorption. There was a significant difference in albumin, another indicator of protein metabolism, between the L and H groups at the time of weaning. However, considering the blood albumin concentration ranges during gestation (3.3–4.9 mg/dL) and suckling (3.5–4.9 mg/dL) according to Verheyen et al. [36], it can be concluded that the blood albumin concentration of the sows in our study was within appropriate levels. Nevertheless, because the H group, which consumed twice as much feed during the lactating period as during the gestation period, showed the lowest albumin values at weaning, it was hypothesized that the quantity of BSF meal fed may have influenced sow protein metabolism. To further test this hypothesis, it is necessary to investigate how proteins and amino acids in BSF meal undergo digestion by conducting in vitro digestion tests.

In the analysis of sow blood cell composition, the lymphocyte count was consistently low in the H group in all three blood sampling instances. In general, a low lymphocyte count indicates a weakened immune system. Healthy sows typically have a white blood cell count ranging from 3.1 to 22.3 × 10^9^/L, with lymphocyte percentages between 1.3 and 6.7 × 10^9^/L [37,38]. This indicated that the lymphocyte counts of the sows in this study were within the expected range. However, as sow immunity significantly affects reproductive performance, further comprehensive studies incorporating BSF meals into sow diets are required. Two weeks before farrowing, the PDW was lower in the L group when compared with the C group. PDW provides insight into the range of platelet size distribution, a factor known to be lower in aplastic anemia. Although there is no available information on the typical PDW range in sows, the values observed in this study were considered to be within the expected range.

Although there were no significant differences in the growth performance of piglets after weaning, the FCR from 28 to 35 days of age was lower in the BSF-fed groups than in the C group, suggesting that the substitution rate between animal protein and BSF meal in the pre-weaning diet may not be adequate. Unlike fish meal and plasma proteins, BSF meal contains chitin, which may hinder protein digestion and absorption in weanling piglets compared with other animal proteins. Acid mammalian chitinase, the enzyme that degrades chitin in insect meals, is secreted from the stomach tissue of piglets after suckling [39]. While piglets are not completely unable to digest chitin, there may be an upper limit to the amount of chitin they can digest. As mentioned in a previous study [40], it may be necessary to remove insect chitin when feeding BSF meals to piglets during weaning. This could involve the addition of an approach that effectively eliminates chitin from insect meals to the current processing method. 

In the internal organs of piglets at 28 days of age, the weight of the kidneys was significantly lower in the H group than in the C and L groups. Previous studies reported that piglet kidney size is affected by the amount of protein ingested [41]. Although not measured in this experiment, the weaned litter size of the H group was the highest; therefore, milk ingestion per suckling piglet was probably the lowest. The amount of digestible protein they could utilize from the pre-weaning diets provided for 7 days prior to weaning was also likely lower for the BSF powder than for regular animal protein. This may have caused the differences in kidney size. Post-weaning diets contained up to 8% BSF meal, and substituting most animal proteins with BSF powder did not affect body weight gain. However, it is still unclear whether the 8% BSF meal content in post-weaning diets is sufficient or whether feeding high concentrations of BSF meal from the early weaning period changed their intestinal environment and adapted their microbiota to produce short-chain fatty acids from the undigested substances in the BSF meal, thereby enabling piglets to gain metabolizable energy. Moreover, it has been suggested that BSF meal may have a positive impact on the immunity of various animal species [11,12]. As such, further investigation is warranted to assess the implications of a prolonged dietary inclusion of BSF meal on the nutritional physiology of piglets.

Regarding the blood cell composition of piglets, significant differences were observed in MCV and MCH at 63 days of age. However, both values were within normal ranges for piglets [42]. Although there were no significant differences in any of the blood properties among the groups, the HDL-C levels at 28 days of age tended to be higher in the H group. This value (88.2 ± 6.0) was above the normal range (29.7–74.5) reported in a previous study [42]. Several studies have reported that BSF oil contains the highest amount of lauric acid [43,44,45,46,47], and its fatty acid composition differs significantly from that of soybean oil. Moreover, research has demonstrated a significant increase in lauric acid in crayfish with the substitution of fishmeal with BSF meal [48]. This noteworthy effect might extend to pork, warranting an in-depth investigation into the long-term impact of this dietary choice on pork quality. The saturated fatty acids from BSF in the pre-weaning diets were likely used for cholesterol synthesis in the livers of piglets [49,50], possibly resulting in higher blood cholesterol levels in the H group. Further research is needed to determine how increased blood cholesterol levels during the suckling period affect the subsequent growth of piglets. It is important to note that BSF meal production currently faces limitations, with its predominant usage confined to aquatic and poultry diets. However, to enhance the sustainability of swine production, rigorous large-scale and long-term feeding trials must be conducted to evaluate the suitability of BSF meal in swine diets.

## 5. Conclusions

This study investigated the substitution of animal protein in the dietary regimen of sows and piglets with BSF meal. The study findings revealed no adverse effects on the reproductive efficiency or hematological properties of sows. Furthermore, the trial outcome highlighted that the complete replacement of animal proteins with BSF meal in pre-weaning diets may induce a reduction in the body weight gain of piglets. However, substituting a predominant fraction of the animal protein in the post-weaning diets with BSF was not detrimental to the piglets. These results support the feasibility of using BSF meal as a substitute for animal protein, soybean meal, and soybean oil in sow and post-weaning diets, thereby enhancing their sustainability. Notably, even in pre-weaning diets, half of the animal protein can be effectively substituted with no notable deterioration. Potential advancements in BSF meal processing present an imminent prospect of completely replacing animal-derived proteins with BSF meals. However, practical implementation should include expanding feeding trials to fattening pigs and continuing insect meal feeding in piglets after weaning, accompanied by a thorough analysis of their effects on the nutrition and physiology of piglets.

## Figures and Tables

**Table 1 animals-13-03410-t001:** Experimental feed ingredients and chemical composition.

Ingredient (%)	Sow			Pre-Weaning			Post-Weaning		
	C	L	H	C	L	H	C	L	H
Corn	62.9	63.4	63.3	35.0	34.8	32.0	48.5	50.0	51.5
Soybean meal	10.0	9.4	7.1	5.0	4.2	2.8	22.0	20.8	19.5
Breadcrumbs	-	-	-	23.0	25.0	30.0	10.0	10.0	10.0
Wheat flour	-	-	-	5.0	5.0	5.0	5.0	5.2	5.5
Wheat bran	0.0	0.0	2.5	-	-	-	-	-	-
Corn gluten feed	4.4	4.5	4.5	-	-	-	-	-	-
Dried Distillers Grains Soluble	4.2	4.4	4.4	-	-	-	-	-	-
Rice bran	7.7	7.8	7.8	-	-	-	0.1	0.1	0.1
Black soldier fly larvae meal (defatted)	0.0	2.3	4.6	0.0	12.5	25.0	0.0	4.0	8.0
Fish meal (CP60%)	-	-	-	10.0	5.0	0.0	4.0	2.0	0.0
Fish meal (CP65%)	4.6	2.3	0.0	-	-	-	-	-	-
Skimmed milk	-	-	-	5.0	2.5	0.0	4.0	2.0	0.0
Whey protein concentrate (CP80%)	-	-	-	5.0	2.5	0.0	-	-	-
Spray dried porcine plasma	-	-	-	5.0	2.5	0.0	3.0	2.5	2.0
Corn oil	0.5	0.1	0.0	4.0	3.0	2.0	-	-	-
Animal fat	2.5	2.6	2.6	-	-	-	-	-	-
Tricalcium phosphate	1.5	1.5	1.5	2.0	2.0	2.0	2.0	2.0	2.0
Calcium carbonate	0.5	0.5	0.5	-	-	-	-	-	-
Salt	0.3	0.3	0.3	0.5	0.5	0.5	0.5	0.5	0.5
Vitamin mineral mix ^1^	0.2	0.2	0.2	0.5	0.5	0.7	0.8	0.7	0.7
Lysine hydrochloride	0.7	0.8	0.8	-	-	-	0.1	0.2	0.2
L-Threonine	0.0	0.0	0.0	-	-	-	-	-	-

^1^ Vitamin and mineral mix: water (3.4%); Na (300.00 g/kg); Fe (8.20 g/kg); Zn (2.20 g/kg); Mn (10.00 g/kg); Cu (0.62 g/kg); choline (30.00 g/kg); vitamin A (3,000,000 IU/kg); vitamin B1 (0.14 g/kg); vitamin B2 (0.35 g/kg); vitamin B3 (1.40 g/kg); vitamin B5 (0.50 g/kg); vitamin B6 (0.25 g/kg); vitamin B7 (0.50 g/kg); vitamin B9 (0.025 g/kg); vitamin B12 (0.0005 g/kg); vitamin D (600,000 IU/kg); vitamin E (596 IU/kg). C, control group; L, group supplemented with low concentrations of black soldier fly larval meal; H, group supplemented with high concentrations of black soldier fly larval meal; CP: crude protein.

**Table 2 animals-13-03410-t002:** Proximal and chemical composition of the experimental feed and black soldier fly (BSF) meal.

Composition (%)	Sow			Pre-Weaning			Post-Weaning			BSF (Defatted)
	C	L	H	C	L	H	C	L	H	
Crude protein ^1,2^	16.33	16.20	15.28	23.30	23.00	22.62	19.58	19.54	19.46	55.23
Crude fat ^1^	6.12	5.90	5.86	6.99	7.06	7.16	6.16	5.91	5.67	11.06
Crude fiber ^1^	2.93	3.06	3.33	1.13	2.00	2.80	2.18	2.43	2.68	8.28
Crude ash ^1^	5.41	5.17	5.08	3.66	3.75	3.85	3.39	3.20	3.00	10.12
NFE ^3^	69.21	69.67	70.45	64.92	64.19	63.57	68.69	68.92	69.19	15.31
Ca ^1^	0.88	0.87	0.80	1.19	1.30	1.40	1.01	0.98	0.96	2.90
P ^1^	0.84	0.80	0.76	0.93	0.82	0.72	0.83	0.77	0.72	1.05
Na ^1^	0.23	0.19	0.19	0.26	0.26	0.25	0.26	0.24	0.22	0.08
Lysine ^1^	1.31	1.37	1.30	1.36	1.23	1.10	1.17	1.23	1.20	3.35
Methionine ^1^	0.30	0.29	0.27	0.39	0.40	0.41	0.33	0.32	0.31	1.14
Threonine ^1^	0.63	0.60	0.57	0.91	0.88	0.84	0.73	0.73	0.73	2.32
Chitin ^1,3^	-	0.15 ^3^	0.30 ^3^	-	0.82 ^3^	1.64 ^3^	-	0.26 ^3^	0.52 ^3^	6.54 ^1^
ME (Mcal/kg) ^3,4^	4.10	4.10	4.09	4.27	4.24	4.21	4.21	4.20	4.18	3.80

^1^ Analyzed data. ^2^ Crude protein = analyzed nitrogen value × (6.25 × non-insects ingredients content + 5.60 × BSF meal content). ^3^ Calculated data. ^4^ ME = (Crude protein × 3.84 + Crude fat × 9.33 + NFE × 4.2)/100. C, control group; L, group supplemented with low concentrations of black soldier fly larval meal; H, group supplemented with high concentrations of black soldier fly larval meal; NFE, nitrogen-free extract; ME, metabolizable energy; BSF, defatted black soldier fly larvae meal.

**Table 3 animals-13-03410-t003:** Reproductive performance of the sows (*n* = 10).

Items	C	L	H	SEM	*p*-Value
Birth litter size ^†^	9.9 ± 1.3	8.8 ± 1.2	12.3 ± 0.8	0.68	0.064
Weaning litter size ^†^	9.0 ± 1.1	7.9 ± 1.3	10.9 ± 0.7	0.63	0.116
Weaning rate ^‡^	92.4 ± 2.1	89.4 ± 6.2	90.4 ± 4.8	2.60	0.833
Body weight at birth (g) ^‡^	1227.3 ± 66.7	1449.4 ± 98.2	1207.8 ± 54.9	46.75	0.162
Body weight at 21 d (g) ^‡^	6648.1 ± 449.8	7538.2 ± 527.8	5671.8 ± 233.8	274.65	0.100
Body weight at weaning (g) ^‡^	9113.1 ± 557.4	9956.3 ± 705.8	7685.4 ± 277.9	349.22	0.205
Body weight gain (21–28 d, g) ^‡^	2465.0 ± 162.3	2418.1 ± 224.0	2013.7 ± 110.9	102.96	0.518
Body weight gain (g) ^‡^	7925.0 ± 517.6	8498.1 ± 627.4	6439.8 ± 277.4	319.91	0.186

Data: mean ± standard error. ^†^ Analysis of covariance was performed using the parity number of sows as a covariate. ^‡^ Analysis of covariance was performed using the birth litter size as a covariate. C, control group; L, group supplemented with low concentrations of black soldier fly larval meal; H, group supplemented with high concentrations of black soldier fly larval meal; SEM, standard error of the mean.

**Table 4 animals-13-03410-t004:** Growth performance of the piglets (*n* = 5).

Items	C	L	H	SEM	*p*-Value
Body weight at 28 d (kg) ^†^	8.73 ± 0.38	9.65 ± 0.67	8.50 ± 0.31	0.24	0.364
Body weight at 35 d (kg) ^‡^	10.70 ± 0.43	11.20 ± 0.67	9.79 ± 0.24	0.29	0.241
Body weight at 63 d (kg) ^‡^	27.09 ± 1.46	28.91 ± 2.27	25.34 ± 1.43	1.04	0.809
Body weight gain (28–35 d, kg) ^‡^	1.97 ± 0.26	1.53 ± 0.30	1.29 ± 0.20	0.14	0.241
Body weight gain (35–63 d, kg) ^‡^	16.39 ± 1.49	17.75 ± 1.81	15.54 ± 1.55	0.92	0.786
Body weight gain (28–64 d, kg) ^‡^	18.36 ± 1.66	19.20 ± 1.99	16.83 ± 1.70	1.00	0.809
Feed intake (28–35 d, kg) ^‡^	3.44 ± 0.17	2.96 ± 0.34	2.81 ± 0.09	0.14	0.171
Feed intake (35–63 d, kg) ^‡^	33.38 ± 1.10	33.39 ± 3.71	30.69 ± 3.19	1.58	0.255
Feed intake (28–63 d, kg) ^‡^	36.81 ± 1.21	36.34 ± 3.82	33.50 ± 3.23	1.64	0.303
Feed conversion rate (28–35 d) ^‡^	1.94 ± 0.38	2.23 ± 0.41	2.45 ± 0.47	0.21	0.347
Feed conversion rate (35–63 d) ^‡^	2.11 ± 0.20	1.99 ± 0.32	2.01 ± 0.17	0.12	0.830
Feed conversion rate (28–63 d) ^‡^	2.08 ± 0.21	1.99 ± 0.29	2.02 ± 0.15	0.12	0.899

Data: mean ± standard error. ^†^ Analysis of covariance was performed using the sex of piglets as a covariate. ^‡^ Analysis of covariance was performed by using the body weight at 28 d and the sex of piglets as a covariate. C, control group; L, group supplemented with low concentrations of black soldier fly larval meal; H, group supplemented with high concentrations of black soldier fly larval meal; SEM, standard error of the mean.

**Table 5 animals-13-03410-t005:** Selected hematological indices of the sows (*n* = 10).

Items	C	L	H	SEM	*p*-Value
2 months after AI (60 d)					
RBC (×10^12^)	7.1 ± 0.3	6.9 ± 0.2	6.8 ± 0.3	0.14	0.74
WBC (×10⁹/L)	14.4 ± 1.4	13.9 ± 0.9	12.4 ± 1.0	0.65	0.47
Lymph (×10⁹/L)	6.2 ± 0.8 ^ab^	6.9 ± 0.5 ^a^	4.8 ± 0.4 ^b^	0.36	0.04 *
Mon (×10⁹/L)	0.8 ± 0.2	0.5 ± 0.1	0.5 ± 0.1	0.07	0.12
Gran (×10⁹/L)	7.4 ± 1.1	6.5 ± 0.6	6.0 ± 0.7	0.44	0.51
Lymph (%)	44.2 ± 3.9	50.4 ± 2.2	43.4 ± 3.8	1.88	0.25
Mon (%)	5.0 ± 0.4	3.8 ± 0.3	4.2 ± 0.4	0.22	0.07
Gran (%)	50.8 ± 3.8	45.9 ± 2.0	52.4 ± 3.7	1.79	0.30
PLT (×10⁹/L)	161.1 ± 15.0	148.6 ± 14.7	185.0 ± 17.0	9.13	0.28
HGB (g/L)	141.6 ± 4.4	136.6 ± 2.8	136.0 ± 6.1	2.62	0.66
HCT (%)	45.1 ± 1.4	43.5 ± 0.9	43.6 ± 1.9	0.83	0.72
MCV (fL)	64.0 ± 0.8	62.9 ± 0.9	64.0 ± 0.8	0.47	0.42
MCH (pg)	20.0 ± 0.3	19.7 ± 0.2	19.9 ± 0.2	0.14	0.66
MCHC (g/L)	313.7 ± 1.7	314.0 ± 1.3	311.2 ± 1.9	0.94	0.24
RDW (%)	15.6 ± 0.3	15.4 ± 0.2	15.6 ± 0.3	0.15	0.82
MPV (fL)	9.5 ± 0.3	9.2 ± 0.3	9.5 ± 0.3	0.15	0.59
PDW (%)	16.9 ± 0.2	16.5 ± 0.1	16.8 ± 0.2	0.11	0.41
PCT (%)	0.2 ± 0.0	0.1 ± 0.0	0.2 ± 0.0	0.01	0.25
2 weeks before farrowing (100 d)					
RBC (×10^12^)	6.8 ± 0.4	6.7 ± 0.2	6.4 ± 0.2	0.15	0.42
WBC (×10⁹/L)	13.0 ± 1.0	13.0 ± 1.2	10.5 ± 0.8	0.60	0.15
Lymph (×10⁹/L)	5.5 ± 0.6	6.1 ± 0.6	4.2 ± 0.3	0.33	0.06
Mon (×10⁹/L)	0.6 ± 0.1	0.6 ± 0.1	0.4 ± 0.0	0.04	0.09
Gran (×10⁹/L)	6.4 ± 0.8	6.7 ± 0.9	4.9 ± 0.3	0.43	0.21
Lymph (%)	44.3 ± 1.5	45.5 ± 2.5	44.0 ± 1.9	1.19	0.88
Mon (%)	5.0 ± 0.3	4.7 ± 0.4	4.4 ± 0.6	0.24	0.66
Gran (%)	50.8 ± 1.6	49.8 ± 2.4	51.6 ± 2.1	1.20	0.84
PLT (×10⁹/L)	111.9 ± 18.1	140.7 ± 22.8	154.7 ± 23.1	12.4	0.29
HGB (g/L)	138.5 ± 6.7	131.4 ± 2.1	128.2 ± 4.0	2.70	0.28
HCT (%)	43.8 ± 2.0	42.3 ± 0.7	40.7 ± 1.1	0.82	0.30
MCV (fL)	64.6 ± 0.9	63.7 ± 1.0	64.3 ± 0.9	0.53	0.66
MCH (pg)	20.3 ± 0.2	19.7 ± 0.3	20.1 ± 0.3	0.16	0.33
MCHC (g/L)	315.4 ± 1.8	310.6 ± 2.1	312.5 ± 2.2	1.20	0.23
RDW (%)	15.6 ± 0.3	16.4 ± 0.4	15.9 ± 0.3	0.19	0.19
MPV (fL)	9.3 ± 0.2	9.4 ± 0.2	9.5 ± 0.3	0.14	0.85
PDW (%)	17.2 ± 0.2 ^a^	16.5 ± 0.1 ^b^	16.6 ± 0.1 ^ab^	0.10	0.02 *
PCT (%)	0.1 ± 0.0	0.1 ± 0.0	0.1 ± 0.0	0.01	0.26
Weaning (142 d)					
RBC (×10^12^)	5.8 ± 0.3	6.1 ± 0.2	5.9 ± 0.1	0.12	0.61
WBC (×10⁹/L)	14.4 ± 0.7	13.9 ± 1.4	13.8 ± 0.9	0.59	0.91
Lymph (×10⁹/L)	4.2 ± 0.5	4.6 ± 0.6	3.6 ± 0.4	0.31	0.40
Mon (×10⁹/L)	0.6 ± 0.1	0.7 ± 0.1	0.6 ± 0.1	0.04	0.64
Gran (×10⁹/L)	9.1 ± 0.4	8.6 ± 1.0	8.9 ± 0.8	0.46	0.94
Lymph (%)	30.0 ± 2.1	33.4 ± 2.4	28.3 ± 2.8	1.47	0.35
Mon (%)	4.2 ± 0.4	5.0 ± 0.4	4.5 ± 0.5	0.26	0.53
Gran (%)	65.8 ± 1.9	61.6 ± 2.6	67.2 ± 2.8	1.51	0.29
PLT (×10⁹/L)	166.2 ± 20.3	191.9 ± 29.5	165.4 ± 27.6	14.7	0.73
HGB (g/L)	104.8 ± 11.7	120.7 ± 4.3	118.4 ± 2.3	4.27	0.29
HCT (%)	36.8 ± 1.6	38.7 ± 1.2	37.5 ± 0.6	0.69	0.57
MCV (fL)	63.8 ± 0.8	63.7 ± 0.7	64.4 ± 0.6	0.40	0.53
MCH (pg)	20.0 ± 0.3	19.8 ± 0.2	20.0 ± 0.2	0.13	0.51
MCHC (g/L)	313.9 ± 2.2	311.1 ± 2.1	312.1 ± 2.2	1.23	0.69
RDW (%)	15.4 ± 0.3	16.2 ± 0.4	15.7 ± 0.4	0.20	0.22
MPV (fL)	8.8 ± 0.2	9.2 ± 0.2	8.9 ± 0.2	0.12	0.51
PDW (%)	17.0 ± 0.2	16.6 ± 0.2	17.0 ± 0.2	0.13	0.38
PCT (%)	0.1 ± 0.0	0.2 ± 0.0	0.2 ± 0.1	0.03	0.63

Data: mean ± standard error. Analysis of covariance was performed using the parity number of sows as a covariate. The *p*-value, denoting significance, is indicated with an asterisk. The values within a row with different superscripts represent statistical differences in multiple comparisons (*p* < 0.05). C, control group; L, group supplemented with low concentrations of black soldier fly larval meal; H, group supplemented with high concentrations of black soldier fly larval meal; SEM, standard error of the mean; RBC, red blood cell; WBC, white blood cell; Lymph, lymphocyte; Mon, monocyte; Gran, granulocyte; PLT, platelet; HGB, hemoglobin; HCT, hematocrit; MCV, mean corpuscular volume; MCH, mean corpuscular hemoglobin; MCHC, mean corpuscular hemoglobin concentration; RDW, red cell distribution width; MPV, mean platelet volume; PDW, platelet distribution width; PCT, platelet count.

**Table 6 animals-13-03410-t006:** Selected plasma biochemical parameters of the sows (*n* = 10).

Items	C	L	H	SEM	*p*-Value
2 months after AI (60 d)					
Glu (mg/dL)	84.6 ± 1.8	82.8 ± 2.3	82.7 ± 3.7	1.50	0.851
T-Cho (mg/dL)	75.7 ± 3.1	76.7 ± 1.7	78.0 ± 3.7	1.66	0.865
HDL-c (mg/dL)	22.7 ± 1.8	25.1 ± 1.3	24.9 ± 3.6	1.37	0.681
LDL-c (mg/dL)	53.0 ± 2.3	51.6 ± 1.6	53.1 ± 2.5	1.23	0.760
T-Pro (g/dL)	7.0 ± 0.1	7.1 ± 0.1	7.0 ± 0.2	0.09	0.984
Alb (g/dL)	4.1 ± 0.1	4.4 ± 0.1	4.2 ± 0.1	0.05	0.045 *
2 weeks before farrowing (100 d)					
Glu	90.8 ± 3.3	91.0 ± 2.3	86.0 ± 2.4	1.56	0.210
T-Cho	70.5 ± 3.1	70.8 ± 2.8	68.5 ± 3.4	1.75	0.838
HDL-c	21.1 ± 2.1	20.2 ± 1.6	20.0 ± 2.3	1.12	0.926
LDL-c	49.4 ± 2.0	50.6 ± 1.6	48.5 ± 1.8	1.04	0.736
T-Pro	7.1 ± 0.1	7.2 ± 0.2	6.9 ± 0.2	0.09	0.659
Alb	4.1 ± 0.1	4.2 ± 0.1	4.0 ± 0.1	0.05	0.441
Weaning (142 d)					
Glu	85.4 ± 5.9	84.7 ± 3.3	92.6 ± 6.6	3.11	0.483
T-Cho	84.9 ± 4.8	89.0 ± 7.1	79.2 ± 3.3	3.06	0.451
HDL-c	43.2 ± 4.0	44.5 ± 6.3	36.1 ± 2.7	2.66	0.420
LDL-c	41.7 ± 2.2	44.5 ± 3.2	43.1 ± 2.5	1.49	0.706
T-Pro	6.7 ± 0.1	6.9 ± 0.2	6.7 ± 0.1	0.08	0.472
Alb	4.0 ± 0.1 ^ab^	4.1 ± 0.4 ^a^	3.7 ± 0.1 ^b^	0.07	0.038 *

Data: mean ± standard error. Analysis of covariance was performed using the parity number of sows as a covariate. The *p*-value, denoting significance, is indicated with an asterisk. The values within a row with different superscripts represent statistical differences in multiple comparisons (*p* < 0.05). C, control group; L, group supplemented with low concentrations of black soldier fly larval meal; H, group supplemented with high concentrations of black soldier fly larval meal; SEM, standard error of the mean; Glu, glucose; T-Cho, total cholesterol; HDL-Cho, high-density lipoprotein cholesterol; LDL-Cho, low-density lipoprotein cholesterol; T-Pro, total protein; Alb, albumin.

**Table 7 animals-13-03410-t007:** Selected internal organ weights and lengths of the piglets.

Items	C	L	H	SEM	*p*-Value
28 d (*n* = 10)					
Body weight (kg) ^†^	8.3 ± 0.4 ^ab^	9.3 ± 0.3 ^a^	7.5 ± 0.3 ^b^	0.24	0.032 *
Liver (g/kg) ^‡^	24.1 ± 1.0	22.8 ± 0.5	23.3 ± 1.0	0.48	0.541
Spleen (g/kg) ^‡^	2.1 ± 0.2	2.4 ± 0.2	1.8 ± 0.1	0.12	0.104
Kidney (g/kg) ^‡^	5.7 ± 0.2 ^a^	5.9 ± 0.2 ^a^	5.0 ± 0.2 ^b^	0.12	<0.001 *
Gastrointestinal tract (g/kg) ^‡^	56.1 ± 3.6	47.7 ± 1.5	52.9 ± 3.6	1.83	0.111
Stomach (g/kg) ^‡^	5.4 ± 0.3	4.9 ± 0.2	4.6 ± 0.3	0.16	0.077
Small intestinal weight (g/kg) ^‡^	32.7 ± 3.1	26.2 ± 0.6	32.1 ± 2.9	1.48	0.110
Small intestinal length (cm) ^§^	787.2 ± 52.2	730.4 ± 41.1	688.0 ± 37.1	25.60	0.157
63 d (*n* = 15)					
Body weight (kg) ^‡^	26.3 ± 1.1	29.3 ± 2.0	24.0 ± 1.3	0.89	0.053
Liver (g/kg) ^‡^	26.7 ± 0.6	29.2 ± 2.6	26.1 ± 0.9	0.94	0.324
Spleen (g/kg) ^‡^	1.8 ± 0.2	2.1 ± 0.2	1.8 ± 0.1	0.11	0.532
Kidney (g/kg) ^‡^	5.9 ± 0.3	6.3 ± 0.7	5.5 ± 0.3	0.27	0.456
Gastrointestinal tract (g/kg) ^‡^	97.5 ± 3.1	94.6 ± 8.6	94.7 ± 4.3	3.30	0.918
Stomach (g/kg) ^‡^	7.6 ± 0.2	8.1 ± 0.9	7.7 ± 0.4	0.33	0.769
Small intestinal weight (g/kg) ^‡^	53.5 ± 2.8	57.5 ± 5.3	51.7 ± 2.5	2.14	0.500
Small intestinal length (cm) ^§^	1380.8 ± 57.4	1415.5 ± 23.6	1318.4 ± 46.3	25.92	0.271

^†^ Analysis of covariance was performed using the litter size at the weaning as a covariate. ^‡^ Analysis of covariance was performed using the sex of the piglets as a covariate. ^§^ Analysis of covariance was performed using the body weight and sex of the piglets as covariates. The *p*-value, denoting significance, is indicated with an asterisk. The values within a row with different superscripts represent the statistical differences in multiple comparisons (*p* < 0.05). C, control group; L, group supplemented with low concentrations of black soldier fly larval meal; H, group supplemented with high concentrations of black soldier fly larval meal; SEM, standard error of the mean.

**Table 8 animals-13-03410-t008:** Intestinal histomorphometry of the piglets.

Items	C	L	H	SEM	*p*-Value
28 d (*n* = 10)					
Duodenum					
Villi height	510.6 ± 58.6	540.7 ± 54.7	543.9 ± 31.7	22.12	0.654
Crypt depth	196.8 ± 13.7 ^a^	264.8 ± 10.3 ^b^	261.2 ± 13.5 ^b^	6.70	0.002 *
Villi/Crypt	2.6 ± 0.2	2.0 ± 0.2	2.1 ± 0.1	0.09	0.226
Jejunum					
Villi height	507.8 ± 34.1	480.7 ± 19.0	537.0 ± 34.1	13.88	0.609
Crypt depth	194.7 ± 15.1	184.6 ± 9.4	210.5 ± 14.6	6.01	0.283
Villi/Crypt	2.7 ± 0.2	2.7 ± 0.2	2.7 ± 0.3	0.10	0.919
Ileum					
Villi height	330.2 ± 16.3	328.4 ± 21.0	324.6 ± 18.6	8.29	0.217
Crypt depth	172.0 ± 14.2	167.4 ± 17.0	159.6 ± 5.4	5.46	0.201
Villi/Crypt	2.0 ± 0.1	2.1 ± 0.1	2.0 ± 0.1	0.06	0.873
63 d (*n* = 15)					
Duodenum					
Villi height	375.3 ± 13.6	379.9 ± 20.2	426.8 ± 31.6	13.46	0.165
Crypt depth	378.7 ± 20.8	368.3 ± 14.4	384.1 ± 18.3	10.23	0.842
Villi/Crypt	1.0 ± 0.1	1.1 ± 0.1	1.1 ± 0.1	0.04	0.487
Jejunum					
Villi height	368.5 ± 20.8 ^a^	421.4 ± 19.0 ^ab^	429.0 ± 26.7 ^b^	13.29	0.051
Crypt depth	255.4 ± 13.0	259.7 ± 6.7	282.3 ± 14.4	6.92	0.396
Villi/Crypt	1.5 ± 0.1	1.6 ± 0.1	1.6 ± 0.1	0.06	0.508
Ileum					
Villi height	396.4 ± 27.9	309.7 ± 23.8	326.4 ± 21.1	14.91	0.036 *
Crypt depth	251.5 ± 10.0	241.3 ± 6.4	243.7 ± 10.8	5.26	0.718
Villi/Crypt	1.6 ± 0.1	1.3 ± 0.1	1.4 ± 0.1	0.07	0.223

Data: mean ± standard error. Analysis of covariance was performed using the body weight and sex of the piglets as covariates. The *p*-value, denoting significance, is indicated with an asterisk. The values within a row with different superscripts represent the statistical differences in multiple comparisons (*p* < 0.05). C: control group; L: group supplemented with low concentrations of black soldier fly larval meal; H: group supplemented with high concentrations of black soldier fly larval meal; SEM: standard error of the mean.

**Table 9 animals-13-03410-t009:** Selected hematological indices of the piglets.

Items	C	L	H	SEM	*p*-Value
28 d (*n* = 10)					
RBC (×10^12^)	6.1 ± 0.2	6.2 ± 0.2	6.3 ± 0.1	0.11	0.843
WBC (×10⁹/L)	11.4 ± 2.6	11.7 ± 1.3	6.7 ± 1.0	1.07	0.090
Lymph (×10⁹/L)	5.8 ± 1.5	3.9 ± 0.8	3.4 ± 0.5	4.07	0.292
Mon (×10⁹/L)	1.2 ± 0.3	1.2 ± 0.3	0.5 ± 0.2	1.67	0.155
Gran (×10⁹/L)	7.2 ± 3.0	6.6 ± 1.5	2.5 ± 0.4	3.86	0.162
Lymph (%)	47.9 ± 7.9	38.8 ± 8.1	53.4 ± 2.7	4.07	0.328
Mon (%)	12.5 ± 5.3	9.2 ± 1.6	7.7 ± 1.9	1.67	0.564
Gran (%)	43.9 ± 7.6	52.0 ± 7.8	38.9 ± 2.0	3.86	0.363
PLT (×10⁹/L)	562.8 ± 45.7	414.2 ± 46.7	541.9 ± 21.5	25.27	0.033 *
HGB (g/L)	93.3 ± 4.1	93.6 ± 7.5	92.9 ± 3.8	3.01	0.817
HCT (%)	31.2 ± 1.3	32.1 ± 2.5	31.8 ± 1.3	0.99	0.956
MCV (fL)	51.7 ± 2.4	51.8 ± 3.2	50.6 ± 2.1	1.45	0.792
MCH (pg)	15.4 ± 0.8	15.0 ± 1.0	14.7 ± 0.6	0.45	0.589
MCHC (g/L)	298.1 ± 3.0	287.7 ± 2.6	291.8 ± 2.8	1.77	0.059
RDW (%)	22.1 ± 1.7	22.2 ± 1.7	22.9 ± 1.1	0.84	0.627
MPV (fL)	7.1 ± 0.2	7.5 ± 0.4	6.6 ± 0.1	0.16	0.053
PDW (%)	15.8 ± 0.2	29.3 ± 14.0	15.3 ± 0.2	4.65	0.438
PCT (%)	0.392 ± 0.033	0.300 ± 0.031	0.357 ± 0.01	0.02	0.094
63 d (*n* = 15)					
RBC (×10^12^)	6.3 ± 0.2	6.8 ± 0.2	6.3 ± 0.2	0.13	0.268
WBC (×10⁹/L)	13.0 ± 0.8	15.5 ± 0.9	13.9 ± 0.8	0.49	0.104
Lymph (×10⁹/L)	7.1 ± 0.5	7.8 ± 0.5	7.1 ± 0.5	0.27	0.548
Mon (×10⁹/L)	0.7 ± 0.1	0.9 ± 0.1	0.7 ± 0.1	0.05	0.195
Gran (×10⁹/L)	5.3 ± 0.5	6.8 ± 0.5	6.0 ± 0.4	0.28	0.063
Lymph (%)	54.9 ± 2.2	50.4 ± 1.8	51.5 ± 1.6	1.11	0.247
Mon (%)	5.2 ± 0.3	5.6 ± 0.5	5.2 ± 0.3	0.22	0.652
Gran (%)	40.0 ± 2.1	43.9 ± 1.7	43.3 ± 1.5	1.05	0.266
PLT (×10⁹/L)	404.6 ± 30.9	671.5 ± 199.1	735.1 ± 209.7	97.15	0.348
HGB (g/L)	103.0 ± 5.2	103.3 ± 54.0	111.9 ± 2.2	2.61	0.278
HCT (%)	34.1 ± 1.7	34.0 ± 1.9	36.3 ± 1.0	0.90	0.517
MCV (fL)	54.2 ± 2.1 ^ab^	50.2 ± 2.2 ^a^	57.5 ± 0.6 ^b^	1.11	0.017 *
MCH (pg)	16.3 ± 0.6 ^ab^	15.2 ± 0.7 ^a^	17.8 ± 0.4 ^b^	0.37	0.009 *
MCHC (g/L)	301.3 ± 1.6	304.9 ± 5.2	309.4 ± 5.7	2.61	0.457
RDW (%)	19.8 ± 0.8	22.1 ± 1.4	20.8 ± 1.8	0.79	0.504
MPV (fL)	7.6 ± 0.3	7.7 ± 0.5	8.8 ± 0.6	0.28	0.191
PDW (%)	15.5 ± 0.2	15.2 ± 0.2	15.5 ± 0.1	0.10	0.421
PCT (%)	0.304 ± 0.020	0.347 ± 0.023	0.339 ± 0.024	0.01	0.244

Data: mean ± standard error. Analysis of covariance was performed using the sex of the piglets as a covariate. The *p*-value, denoting significance, is indicated with an asterisk. The values within a row with different superscripts represent the statistical differences in multiple comparisons (*p* < 0.05). C, control group; L, group supplemented with low concentrations of black soldier fly larval meal; H, group supplemented with high concentrations of black soldier fly larval meal; SEM, standard error of the mean; RBC, red blood cell; WBC, white blood cell; Lymph, lymphocyte; Mon, monocyte; Gran, granulocyte; PLT, platelet; HGB, hemoglobin; HCT, hematocrit; MCV, mean corpuscular volume; MCH, mean corpuscular hemoglobin; MCHC, mean corpuscular hemoglobin concentration; RDW, red cell distribution width; MPV, mean platelet volume; PDW, platelet distribution width; PCT, platelet count.

**Table 10 animals-13-03410-t010:** Selected plasma biochemical parameters of the piglets.

Items	C	L	H	SEM	*p*-Value
28 d (*n* = 10)					
Glu	144.0 ± 9.5	129.3 ± 6.7	128.5 ± 4.8	4.28	0.345
T-Cho	120.3 ± 12.3	118.2 ± 10.8	152.6 ± 14.4	7.60	0.169
HDL-c	70.2 ± 4.3	61.6 ± 5.4	88.2 ± 6.0	3.58	0.051
LDL-c	50.1 ± 9.6	56.6 ± 6.5	64.4 ± 9.7	4.98	0.388
T-Pro	4.1 ± 0.2	4.4 ± 0.2	4.5 ± 0.1	0.10	0.250
Alb	3.0 ± 0.2	3.0 ± 0.2	2.9 ± 0.1	0.11	0.546
63 d (*n* = 15)					
Glu	113.4 ± 3.2	120.9 ± 5.2	111.9 ± 3.5	2.37	0.164
T-Cho	73.6 ± 2.7	77.3 ± 2.4	80.7 ± 3.6	1.72	0.237
HDL-c	31.7 ± 1.9	32.0 ± 2.3	37.6 ± 3.3	1.51	0.139
LDL-c	41.9 ± 1.5	45.3 ± 1.1	43.1 ± 1.3	0.76	0.139
T-Pro	5.0 ± 0.2	4.9 ± 0.1	4.7 ± 0.1	0.07	0.372
Alb	3.0 ± 0.1	3.0 ± 0.1	2.9 ± 0.1	0.07	0.734

Data: mean ± standard error. Analysis of covariance was performed using the body weight and sex of the piglets as covariates. C, control group; L, group supplemented with low concentrations of black soldier fly larval meal; H, group supplemented with high concentrations of black soldier fly larval meal; SEM, standard error of the mean; Glu, glucose; T-Cho, total cholesterol; HDL-Cho, high-density lipoprotein cholesterol; LDL-Cho, low-density lipoprotein cholesterol; T-Pro, total protein; Alb, albumin.

## Data Availability

The raw data used in this study are available from the corresponding author upon reasonable request.

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
