# Peer review of "Sustenance Trial to Analyze the Effects of Black Soldier Fly Larvae Meal on the Reproductive Efficiency of Sows and the Hematological Properties of Suckling and Weaning Piglets"

_animals, 2023, doi:10.3390/ani13213410_

Round 1
Reviewer 1 Report
Comments and Suggestions for Authors
Helmetia insect meals constitute a promising additive in feeding rations of productive animals, mainly aquatic and poultry. The current study evaluated the addition of Helmetia in pig diets, presenting interesting and valuable data. It a well designed study, however there are some major issues that have to be clarified and better discussed.
A major issue of the study is the limited sample size. The authors mention it in the end of the discussion, but the have also to provide a good explanation and justification in the scope or Materials and Methods as well.
In the results, to more clearly indicate statistical significances in the all Tables, apart from the p-values, significant values would be better indicated with an asterisk or bold.
In 2.1, how were the low and high concentrations defined? By substituting a percentage of protein from other source? This is not clear from the Tables and should be better explained
In lines 314-325, apart from increased cholesterol levels, BSF has been also proved to increase fatty acid composition in meat, in aquatic organisms, particularly lauric acid (Alvanou et al. doi: 10.3389/fphys.2023.1156394). This positively affects he quality of the product. Although, fatty acid profiles were not measured, this statement should be added at this point as a future perspective potential.
I generally disagree for the practical point of auditing BSF meals in pig diets relatively low at least so far. The reason that they have been presumably used for fish, aquatic organisms and poultry is the generally lower quantities needed. The authors should add a part in the discussion analysing this practical difficulty
On the other hand, a very promising perspective of the use of insect meals is the immune response. Thus, although perhaps not so practical in terms of required quantities, the potential improvement in post-weaning piglet mortalities, may be a promising perspective that should be also mention in the discussion
Author Response
Responds to the Reviewer’s Comments:
Reviewer 1:
Comments and Suggestions for Authors
Helmetia insect meals constitute a promising additive in feeding rations of productive animals, mainly aquatic and poultry. The current study evaluated the addition of Helmetia in pig diets, presenting interesting and valuable data. It a well designed study, however there are some major issues that have to be clarified and better discussed.
A major issue of the study is the limited sample size. The authors mention it in the end of the discussion, but the have also to provide a good explanation and justification in the scope or Materials and Methods as well.
Thank you for reviewing this manuscript. We also appreciate the positive comments on our research. As you pointed out, we have revised the manuscript to mention sample size at the beginning of the discussion instead of mentioning it at the end of the discussion.
We have also mentioned the need to conduct a rearing study on a large scale for application in swine production (Please see L267-270 and L348-352).
In the results, to more clearly indicate statistical significances in the all Tables, apart from the p-values, significant values would be better indicated with an asterisk or bold.
Thank you for your valuable comment which will help us to improve the readability for the readers. We have asterisked the p-values with significant differences (Please see Table 5-9).
In 2.1, how were the low and high concentrations defined? By substituting a percentage of protein from other source? This is not clear from the Tables and should be better explained
Thank you for your valuable suggestion. We have added sentences regarding the composition of BSF meal in the diets (Please refer to L131-141).
In lines 314-325, apart from increased cholesterol levels, BSF has been also proved to increase fatty acid composition in meat, in aquatic organisms, particularly lauric acid (Alvanou et al. doi: 10.3389/fphys.2023.1156394). This positively affects he quality of the product. Although, fatty acid profiles were not measured, this statement should be added at this point as a future perspective potential.
Thank you for your valuable suggestion. We have added the sentence regarding lauric acid citing the paper you provided (Please see L341-344).
I generally disagree for the practical point of auditing BSF meals in pig diets relatively low at least so far. The reason that they have been presumably used for fish, aquatic organisms and poultry is the generally lower quantities needed. The authors should add a part in the discussion analysing this practical difficulty
Thank you for indicating that. The authors believe that you are correct and that currently insect protein production requires more cost than existing feed ingredients, so there are few applications for swine, which needs a large amount of feed. However, we also believe that the use of BSF meal can help to make the swine industry more sustainable. We have added the above statement to the end of the discussion (Please see L348-352).
On the other hand, a very promising perspective of the use of insect meals is the immune response. Thus, although perhaps not so practical in terms of required quantities, the potential improvement in post-weaning piglet mortalities, may be a promising perspective that should be also mention in the discussion
Thank you for your valuable suggestion. Although we did not observe significant differences in this study due to the sample size of piglets and the short rearing period, previous studies and reviews suggest that insect proteins have a positive effect on animal nutritional physiology, so we have added a sentence to the discussion mentioning this (Please see L330-333).
Reviewer 2 Report
Comments and Suggestions for Authors
Authors could be specify the breed involved in the experiment.
From data reported in Tab1, the following appears to emerge: in the sow diet, the addition of larvae meal, +2.3% in group L and +4.6% in H, seems to replace only fish meal (-2.3% in L and -4,6% in H); in the pre-weaning diet only from animal protein sources (skimmed milk -2.5%, fish meal -5.0%, whey protein concentrate -2.5% and spray dried porcine plasma in L; skimmed milk -5.0%, fish meal -10.0%, whey protein concentrate - 5.0% and spray dried porcine plasma in H) replaced by +12,5% in L and 25.0% in H of larvae meal; in post weaning diet only from fish meal (-2.0% in L and -4.0% in H) and skimmed milk (-2.0% in L and -4.0% in H) replaced by + 4.0% in L and +8.0% in H of larvae meal. There is no evidence of substitution of soybean meal.
Author Response
Responds to the Reviewer’s Comments:
Reviewer 2:
Comments and Suggestions for Authors
Authors could be specify the breed involved in the experiment.
Thank you for reviewing our manuscript. We also thank you for pointing out our missing information. We have added the text about the pig breeds (Please see L113 and L115).
From data reported in Tab1, the following appears to emerge: in the sow diet, the addition of larvae meal, +2.3% in group L and +4.6% in H, seems to replace only fish meal (-2.3% in L and -4,6% in H); in the pre-weaning diet only from animal protein sources (skimmed milk -2.5%, fish meal -5.0%, whey protein concentrate -2.5% and spray dried porcine plasma in L; skimmed milk -5.0%, fish meal -10.0%, whey protein concentrate - 5.0% and spray dried porcine plasma in H) replaced by +12,5% in L and 25.0% in H of larvae meal; in post weaning diet only from fish meal (-2.0% in L and -4.0% in H) and skimmed milk (-2.0% in L and -4.0% in H) replaced by + 4.0% in L and +8.0% in H of larvae meal. There is no evidence of substitution of soybean meal.
Thank you for your valuable comments. As you mentioned, we agree that the word "reduction" is more appropriate for soybean meal, not substitution. Therefore, we changed the purpose of the study to the substitution of fishmeal and animal protein with BSF, and revised the sentence in the conclusion.
Reviewer 3 Report
Comments and Suggestions for Authors
This study used black soldier fly (BSF) larvae meal to replace part of the protein in the feed of sows, suckling and weaned piglets, and found that BSF had no adverse effects on the reproductive efficiency of sows and the hematological characteristics of suckling and weaned piglets. This is a study of great application value. However, the study is not clear about the experimental grouping in Materials and Methods. For example, how about the genetic background of the pigs? How many boars were mated with this female pigs? What is the breed of the boars? How were they distributed among the groups? How many piglets did the 10 female pigs produce? How many sows and piglets were in each group? Did all piglets enter the experiment? Were there any foster piglets? Did any piglets die from lactation to weaning? How many died in each group and what caused them? How were the dead piglets treated in the statistical analysis? Were the effects of maternal effects taken into account in the statistical analysis of growth performances of the piglets? The authors did not specify the background and grouping of the experimental animals, and did not use a mixed model in the statistical analysis to synthesize the various influencing factors. Therefore, the results need to be further verified.
Line 113: what’s the pig’s breed or genetic background? Including the boar. Information on the litter distribution of these gilts and sows is missing. How were 10 female pigs assigned to 3 experimental groups?
Line 116: The male piglets were castrated until 21 days, which is too late. The large age of the piglets at the time of surgery had a greater impact on the piglets. At 1 week of age is better. Also, was there anesthesia for the surgery?
Line 118 and 120: "1.25kg”, “3kg”: Were sows fed the weight of feed for one meal or for the whole day? Highly productive sows can feed up to 8kg or more during lactation. Therefore, information about the genetic background of the female pigs is important.
Line 159: It is recommended that more authoritative statistical software can be used for analysis, such as SPSS or SAS.
Tables: Please add to the table the sample size for each group, i.e., how many pigs were in each group? I'm concerned about the sample size of each group, not the total sample size.
Table 4: what’s means “n=5”? Why was there only growth data for 5 pigs in each group?
Table 5 and 6: n=10? How many sows were in each group?
Author Response
Responds to the Reviewer’s Comments:
Reviewer 3:
Comments and Suggestions for Authors
This study used black soldier fly (BSF) larvae meal to replace part of the protein in the feed of sows, suckling and weaned piglets, and found that BSF had no adverse effects on the reproductive efficiency of sows and the hematological characteristics of suckling and weaned piglets. This is a study of great application value. However, the study is not clear about the experimental grouping in Materials and Methods. For example, how about the genetic background of the pigs? How many boars were mated with this female pigs? What is the breed of the boars? How were they distributed among the groups? How many piglets did the 10 female pigs produce? How many sows and piglets were in each group? Did all piglets enter the experiment? Were there any foster piglets? Did any piglets die from lactation to weaning? How many died in each group and what caused them? How were the dead piglets treated in the statistical analysis? Were the effects of maternal effects taken into account in the statistical analysis of growth performances of the piglets? The authors did not specify the background and grouping of the experimental animals, and did not use a mixed model in the statistical analysis to synthesize the various influencing factors. Therefore, the results need to be further verified.
Thank you for reviewing our manuscript. We also appreciate the positive comment on our study. As you pointed out, we have corrected some omissions and errors in our description of the pig breeds and number of sows. The total number of sows was 30 and 10 sows were assigned to each group (Please see L113-115).
Moreover, we added a sentence regarding the number of piglets farrowed and the number of deaths, see L187-191.
For the statistical treatment of sow reproductive performance, we treated sow fertility as a confounder and performed an analysis of covariance because sows had different reproductive histories; for the statistical treatment of piglet growth performance described in Table 3, we treated the number of piglets farrowed by the sow as a confounder and performed an analysis of covariance.
Line 113: what’s the pig’s breed or genetic background? Including the boar. Information on the litter distribution of these gilts and sows is missing. How were 10 female pigs assigned to 3 experimental groups?
Thank you for your valuable comments. As mentioned above, we made an error in stating the number of sows, and correctly divided the total of 30 sows into 3 groups of 10 sows each. We also added the breed (Please see L113-115).
Line 116: The male piglets were castrated until 21 days, which is too late. The large age of the piglets at the time of surgery had a greater impact on the piglets. At 1 week of age is better. Also, was there anesthesia for the surgery?
Thank you very much for your valuable opinion. We have castrated piglets under Japanese law, but in the future, we will castrate piglets at a younger age, as you suggested. Also, as is the case with Japanese pig farmers, the castration of piglets was done without anesthesia.
Line 118 and 120: "1.25kg”, “3kg”: Were sows fed the weight of feed for one meal or for the whole day? Highly productive sows can feed up to 8kg or more during lactation. Therefore, information about the genetic background of the female pigs is important.
Thank you for your valuable comment. We fed the sows 2.5 kg per day during gestation and 6.0 kg per day during suckling. We revised the description (Please see L120-123).
Line 159: It is recommended that more authoritative statistical software can be used for analysis, such as SPSS or SAS.
Thank you for pointing this out. We retried the statistical process in R and the results obtained were same. However, we revised the description as you suggested (Please see L172).
Tables: Please add to the table the sample size for each group, i.e., how many pigs were in each group? I'm concerned about the sample size of each group, not the total sample size.
Table 4: what’s means “n=5”? Why was there only growth data for 5 pigs in each group?
Table 5 and 6: n=10? How many sows were in each group?
Thank you for pointing this out. We apologize for the confusion caused by our error in stating the total number of sows. All of the n numbers listed in Tables are for each group.
However, for Table 4, it is the number of growing piglet pens. Due to the convenience of the rearing facility, it was not possible to measure the food intake of each piglet individually, so the piglets were reared in herds. Therefore, for Table 4, n = 5, which is the number of pens.
Round 2
Reviewer 1 Report
Comments and Suggestions for Authors
The authors addressed successfully my comments and the manuscript can be published in its current form
Author Response
Thank you for reviewing this manuscript again. We also appreciate the positive comments our manuscript.
Reviewer 3 Report
Comments and Suggestions for Authors
The author's modifications addressed some of my concerns, but there are still data analysis issues as follows:
2.3. Statistical Analysis: The authors state that a one-way ANOVA was used, but covariates were also used. This is clearly incorrect, if there were covariates and the grouping variable was a factor, then the covariate was also a factor and could not be a one-way ANOVA. It should be a multifactor ANOVA, covariate analysis or GLM analysis. If there were random variables, a mixed model analysis should be used.
Table 10: the data of 63d (n=15), should be “mean ± standard error”,but no “±”
Author Response
Reviewer 3:
Comments and Suggestions for Authors
The author's modifications addressed some of my concerns, but there are still data analysis issues as follows:
2.3. Statistical Analysis: The authors state that a one-way ANOVA was used, but covariates were also used. This is clearly incorrect, if there were covariates and the grouping variable was a factor, then the covariate was also a factor and could not be a one-way ANOVA. It should be a multifactor ANOVA, covariate analysis or GLM analysis. If there were random variables, a mixed model analysis should be used.
Thank you for reviewing our manuscript again.
We also appreciate the valuable comments on our Manuscript. Statistical treatment method for piglet growth performance was changed to ANCOVA (Please see L171-184 and Table 4).
Table 10: the data of 63d (n=15), should be “mean ± standard error”,but no “±”
Thank you very much for carefully reviewing our manuscript.
We have added the symbols that were missing from the table (Please see Table 10).